# Composition of Fecal Microbiota in Grazing and Feedlot Angus Beef Cattle

**DOI:** 10.3390/ani11113167

**Published:** 2021-11-05

**Authors:** Zhimin Zhang, Li Yang, Yang He, Xinmao Luo, Shaokang Zhao, Xianbo Jia

**Affiliations:** 1College of Animal Science, Xichang University, Xichang 615000, China; z2463942259@163.com; 2Farm Animal Genetic Resources Exploration and Innovation Key Laboratory of Sichuan Province, Sichuan Agricultural University, Chengdu 611130, China; ylyang1226@163.com (L.Y.); heyang9@outlook.com (Y.H.); lxmrobert@163.com (X.L.); s18884328298@163.com (S.Z.)

**Keywords:** fecal microbioa, grazing cattle, Angus beef, 16S rRNA gene

## Abstract

**Simple Summary:**

This study is to investigate the difference of bovine fecal microbiota between grazing and feedlot Angus cattle. The fecal bacterial community was analyzed by high-throughput sequencing of 16S rRNA gene from six Angus cattle grazed on grassland and six Angus cattle fed on a feedlot. A total of 775 OTUs were taxonomically assigned to bacterial 12 phyla, 19 classes, 25 orders, 54 families, 141 genera, and 145 species. The dominant phyla were Firmicutes and Bacteroidetes. There was similar species richness between grazing and feedlot Angus beef, while species diversity was higher in feedlot Angus beef. The relative abundance of Firmicutes, Cyanobacteria, Elusimicrobia and Patescibacteria was significantly different between grazing and feedlot Angus beef (*p* < 0.05). At the genus level, five microbiotas were significantly different microbiotas between the two groups and all belonged to the Firmicutes phylum. These significant differences in microbiota composition between grazing and feedlot Angus beef may have an impact on the meat quality of Angus beef.

**Abstract:**

This study is to investigate the difference in bovine fecal microbiota between grazing and feedlot Angus cattle. Fecal samples were collected from six Angus cattle grazed on grassland and six Angus cattle fed on a feedlot. The fecal bacterial community was analyzed by high-throughput sequencing of 16S rRNA gene. Sequencing of the V3–V4 region totally produced 1,113,170 effective tages that were computationally clustered into 775 operational taxonomic units (OTUs). These 775 OTUs were taxonomically assigned to bacterial 12 phyla, 19 classes, 25 orders, 54 families, 141 genera, and 145 species. The dominant phyla were Firmicutes and Bacteroidetes. There was similar species richness between grazing and feedlot Angus beef, while higher species diversity was observed in feedlot Angus beef. The relative abundance of Firmicutes, Cyanobacteria, Elusimicrobia and Patescibacteria was significantly different between grazing and feedlot Angus beef (*p* < 0.05). At a genus level, five microbiotas were significantly different between the two groups and all belonged to the Firmicutes phylum. These significant differences in microbiota composition between grazing and feedlot Angus beef may have an impact on the meat quality of Angus beef.

## 1. Introduction

Grazing-fed cattle are free-ranged on pastures with grass as the main feed, while feedlot-fed cattle are raised in feedlots with grains as their main feed. The grass is low in energy but high in fiber, on the contrary, grains are high in energy and easy to digest. Thus, there are significant differences between the two feeding methods in digestibility, growth rate and meat quality. Compared to feedlot-fed beef, grazing-fed beef has lower fatty acid content and higher vitamin content [1]. Because nowadays people are concerned about energy and nutritional balance, consumers are increasingly favoring grazing-fed beef [2]. The proportion of grazing-fed cattle breeding has been gradually increasing as a result of this consumer preference.

Cattle gut microbes interact with their host and participate in the host’s physiological activities such as food digestion, energy metabolism and nutrient absorption, which are closely related to the healthy growth state and methane emissions of the host. Cattle gut microbes are affected by various factors, among which the composition and nutrition of the diet is a key factor in shaping its community structure, abundance and activity. Firmicutes, Bacteroidetes, Proteobacteria, TM7 and Actinobacteria were the main dominant phylum in cattle feces, and their abundance changed significantly with different diets [3]. Firmicutes was the first dominant phylum followed by Bacteroidetes, Proteobacteria, TM7 and Actinobacteria in moderate grain and high grain diets, while Firmicutes was the first dominant phylum followed by TM7, Actinobacteria, Proteobacteria and Bacteroidetes in silage/forage diets [4]. The fecal Firmicute:Bacteroidetes ratio was smaller when beef were fed more than 10% of dietary distiller grain compared with that of a corn diet [5]. There is a tendency to a greater relative abundance of Bacteroidetes but lesser Firmicutes in fecal matter after adding antibiotics or dried yeast to the beef diet [6]. Firmicutes, Bacteroidetes, and Verrucomicrobia were also the dominant phyla in yak, and the Firmicute:Bacteroidetes ratio was significantly decreasing from winter grassland to feedlot feeding [7]. In dairy cattle, diet was the most important parameter to explain fecal microbiota richness, and fecal microbiota of mixed (forage and concentrate) diet was more richness than that of dry forage diet [8].

Angus cattle is well known for its meat quality around the world, and grazing-fed Angus is more favored by consumers. The gut microbiota is significantly different between the grazing- and feedlot-fed beef. This study focuses on understanding the fecal microbiota in grazing- and feedlot-fed beef cattle by 16S rRNA sequencing, and searching for key bacterial differences. This investigation is expected to provide useful information to develop suitable nutritional and management strategies for grazing-fed and feedlot-fed Angus cattle.

## 2. Material and Methods

### 2.1. Fecal Sample Collection

Twelve 14-month-old healthy male Angus cattle (body weight: 405.52 + 8.30 kg) were selected from two nearby farms located in the Liangshan Yi autonomous of Sichuan (China). Six cattle (ANG1~6, group F) were randomly selected from a herd, which was kept in one indoor feedlot and fed with the total mixed ration diet based on corn silage according to the National Research Council (NRC). Another six cattle (ANG7~12, group G) were selected from a herd which grazed in a wild grassland and supplemented only with mineral salt. All cattle were adapted to their current state for more than 3 months before fecal sampling. Samples were collected directly from the rectum of each cattle using a disposable glove, and were immediately frozen in liquid nitrogen in 2 mL cryotubes, and then stored at −80 °C until DNA extraction.

### 2.2. DNA Extraction and Sequencing

Genomic DNA was extracted from the fecal sample using a QIAamp DNA stool Mini Kit (Qiagen, Valencia, CA, USA) according to the manufacturer’s protocol. The variable regions V3–V4 of the 16S rDNA gene were amplified by PCR (98 °C for 2 min, followed by 30 cycles of 98 °C for 30 s, 50 °C for 30 s, 72 °C for 1 min, and 72 °C for 5 min) using bacterial domain-specific primers (338F: 5′-ACTCCTACGGGAGGCAGCAG-3′ and 806R: 5′-GGACTACHVGGGTWTCTAAT-3′). The ends of the primers were added unique barcodes. The PCR products were detected by 1.8% agarose gel electrophoresis, and approximately 450 bp samples were chosen and purified, and built into sequencing libraries. Finally, pair-end sequencing was conducted on an Illumina HiSeq 2500 platform (Illumina, Inc., San Diego, CA, USA).

### 2.3. Bioinformatics Analysis

Raw tags were spliced by merging the overlapping regions between paired-end reads with FLASH V1.2.7 [9], and then they were quality-filtered under specific filtering conditions to obtain high-quality clean tags with Trimmomatic v0.33 [10]. To get effective tags, the chimera sequences were detected and removed by comparing tags with a reference database (Ribosomal Database Program) using UCHIME v4.2 [11]. The effective tags were grouped into operation taxonomic units (OTUs) at 97% sequence identity using Usearch software. The OTUs were annotated based on the taxonomic databases of Sliva (bacteria) and UNITE (fungi).

To get the species classification information of each OTU, we compared the representative sequence of OTU with the microbial reference database. Then we counted the community composition of each sample at each level (phylum, class, order, family, genus, species), and generated species abundance tables at different classification levels by QIIME software [12]. To understand the evolutionary relationship and abundance difference of microorganisms, the species abundance information is returned to the taxonomic system relationship tree of the database by using MEGAN software [13]. The alpha diversity were calculated by Mothur V.1.30 [14] and the beta diversity were analysed for different groups by QIIME for different groups. To analyze the functional differences, we inferred the composition of functional genes in the samples by PICRUSt software [15]. And *t*-test is carried out between different groups and the significance level was set at *p* < 0.05.

## 3. Results

### 3.1. Gene Sequencing Data Summary

The V3–V4 hypervariable regions of 16S rRNA gene were sequenced to fecal microbial communities of 12 Angus cattle. A total of 1,194,599 raw paired-end reads were generated from 12 samples (average: 99,550 ± 26,487, range: 60,156–145,482). And then 1,113,170 effective tages were obtained from 12 samples (average: 92,764 ± 24,560, range: 55,212–134,921) with an average of 411.75 bp per tag after the merging overlapping paired-reads, quality filtering and removing of chimeric sequences. According to the 97% sequence similarity, 775 OTUs were computationally constructed with 687.92 ± 21.48 (range: 659–721) as the mean number of OTUs per sample (Figure 1).

These 775 OTUs were taxonomically assigned to bacterial 12 phyla, 19 classes, 25 orders, 54 families, 141 genera, and 145 species.

### 3.2. Analysis of Bacterial Diversity

The alpha diversity (observed species and Shannon diversity index) was calculated to estimate species richness and diversity in the 12 fecal microbiota samples (Figure 2). There was no statistically significant difference between group F and group G (Figure 2A, *p* = 0.054 of Kruskal-W allis test) in observed species, and a significant difference between group F and group G (Figure 2B, *p* = 0.025 of Kruskal-W allis test) in Shannon diversity index, which indicated that there was a similar species richness between group F and group G, while a higher species diversity in group F.

Principal coordinated analysis (PCoA) based on the binary Jaccard and Bray Curtis methods of beta diversity were further used to analyse compositional differences in fecal microbiota between group F and group G. The animals clustered together according to their particular group, suggesting a compositional shift with respect to community membership and structure between different groups. Both Jaccard and Bray-Curtis distance-based PCoA showed a significant difference between group F and group G (Figure 3, ANOSIM, *p* < 0.05), which indicated that each group hosts its own distinct bacterial community.

### 3.3. Analysis of Bacterial Taxonomy and Function

The taxonomic annotation at the phylum level showed that the common bacterial phyla of fecal microbiota in beef cattle were Firmicutes (with an average relative abundance of 60.61%), Bacteroidetes (28.32%), Verrucomicrobia (7.96%), Spirochaetes (0.91%), Cyanobacteria (0.84%), Proteobacteria (0.70%), Patescibacteria (0.25%), Tenericutes (0.22%), Actinobacteria (0.11%), and Elusimicrobia (0.04%), respectively (Figure 4A). The relative abundance of Firmicutes, Cyanobacteria and Elusimicrobia was significantly higher (*p* < 0.05) in group F compared with group G, while Patescibacteria was significantly lower (*p* < 0.05) in group F. At the genus level, the abundant genus was Ruminococcaceae_UCG-005 (13.30%), Romboutsia (8.82%), Akkermansia (7.96%), Rikenellaceae_RC9_gut_group (7.46%), Prevotellaceae_UCG-003 (4.83%), [Eubacterium]_coprostanoligenes_group (4.82%), Paeniclostridium (4.81%), Ruminococcaceae_UCG-010 (4.63%), Ruminococcaceae_UCG-013 (3.73%) and Bacteroides (3.44%), respectively (Figure 4B). The relative abundance of Romboutsia, Paeniclostridium, [Eubacterium]_coprostanoligenes_group, Ruminococcaceae_UCG-010 and Ruminococcaceae_UCG-013 was significantly higher (*p* < 0.05) in group F compared with group G.

## 4. Discussion

Grazing beef is more popular among consumers for its “natural” provenance and unique beef flavor. Both in dairy cow and beef, grazing was found to affect metabolic properties, thus affecting their skeletal muscle characteristics and content of nutritionally valuable compounds of meat [16]. Grazing Friesian bulls compared with tie-stall housed bulls, Type I %, Type IIA %, Type IIA and IIB fibre areas, capillarization, the activity of citrate synthase, glycogen content and pigmentation were higher in semitendinosus, longissimus dorsi, or supraspinatus muscles [17]. In Charolais steers, an increase mobility in pasture and a grass (vs. maize silage)-based diet contribute to the more oxidative metabolic orientation of muscles [18]. The gut microbiota of the farm animal is one of the important factors affecting the special flavor of meat [19]. In this study, we accordingly investigated the fecal microbiota composition between grazing and feedlot-fed cattle using high-throughput sequencing of 16S rRNA gene.

The composition proportion of fecal microbiota was different among individual, mainly due to diet, age, breed, veterinary drug, and geographic region. Several studies suggested that the community structure of cattle fecal microbiota was greatly affected by diet, especially between forage- and concentrate-based diets [20]. The cluster between forage- and concentrate-based cattle were separate from each other by fecal microbiota, and the difference among individuals of forage-based cattle is greater than that of concentrate-based cattle [4]. In this study, it shown a significant difference between group F and group G according to PCoA. The species diversity was higher in group F than that in group G. The diet composed of forage and grains to increase bacterial diversity in group F.

Firmicutes and Bacteroidetes were the main phyla, accounting for 90% of the cattle fecal bacteria. Firmicutes, the first dominant phylum in cattle fecal samples (>50%), was more abundant in concentrate-based diets than forage-based diets [4]. In this study, the abundance of Firmicutes was significantly different (*p* < 0.05) between the two diet groups with 66.26% in group F and 54.96% in group G. The five microbiotas were significantly different at the genus level between the two groups, all belonging to Firmicutes phylum. Firmicutes are beneficial bacteria in the intestinal tract, which help the host intestinal tract absorb energy from food [21]. The intestinal flora of obese mice has a strong ability to release energy from food, while the intestinal Firmicutes richness of obese mice is significantly higher than that of slim mice [22]. It is also reported that obese people have less Bacteroidetes than slim people, while there are more Firmicutes. After obese people lost weight through diet, obese peoples’ Bacteroidetes increased, while their Firmicutes counts decreased [23]. Here, not only were the Firmicutes of feedlot-fed cattle significantly higher than that of grazing-fed cattle, but also the Bacteroidetes of feedlot-fed cattle was lower than that of grazing-fed cattle (*p* > 0.05). Thus, the Firmicutes/Bacteroidetes ratio was higher in group F than that in group G. In human, the Firmicutes/Bacteroidetes ratio is used as a possible biomarker of gut dysbiosis [24].

Cyanobacteria, Patescibacteria and Elusimicrobia are less common microbes in cow feces, with an abundance of less than 1%. In this study, the relative abundance of Cyanobacteria, Patescibacteria, and Elusimicrobia was 0.84%, 0.25% and 0.04%, respectively. The relative abundance of Cyanobacteria and Elusimicrobia was significantly higher (*p* < 0.05) in group F compared with group G, while Patescibacteria was significantly lower (*p* < 0.05) in group F. Cyanobacteria was used as a source of functional food due to a potential natural alternative to antibiotics, antiviral or antifungal therapies [25]. The increased prevalence of Proteobacteria can be used as a marker for dysbiosis and a potential diagnostic criterion for disease in human gut [26]. Elusimicrobia are gut symbiotic anaerobic bacteria, which produce lactate, acetate, hydrogen and CO_2_ via fermentation [27]. The role of these microbiota in the cattle gut needs to be further studied.

Fecal microbiota transplantation (FMT) is an increasingly popular therapy for the treatment of diseases such as metabolic syndrome, diabetes, Crohn’s disease, Parkinson’s disease, multiple sclerosis, psoriasis, anorexia nervosa or Alzheimer disease in human [28]. Recently, FMT is successfully applied to ameliorate diarrhea and improve growth performance in pre-weaning calves [29]. In this study, the relative abundance of Firmicutes, Cyanobacteria, Elusimicrobia and Patescibacteria was significantly different between the feedlot- and grazing- Angus beef, which may provide the possibility to improve the quality of beef by FMT and microecologic agent.

In conclusion, the current study provides a view of fecal bacterial communities in grazing and feedlot Angus beef by 16S rRNA gene sequences. The 1,113,170 sequences were assigned to bacterial 12 phyla, 19 classes, 25 orders, 54 families, 141 genera, and 145 species. The common bacterial phyla of fecal microbiota were Firmicutes, Bacteroidetes, Verrucomicrobia, Spirochaetes, Cyanobacteria, Proteobacteria, Patescibacteria, Tenericutes, Actinobacteria and Elusimicrobia in order of abundance in Angus beef. There was similar species richness between grazing and feedlot Angus beef, while higher species diversity in feedlot Angus beef. The relative abundance of Firmicutes, Cyanobacteria and Elusimicrobia was significantly higher in feedlot Angus beef compared with grazing Angus beef, while Patescibacteria was significantly lower in feedlot Angus beef. At the genus level, five microbiotas were significantly different between the two groups, all belonging to the Firmicutes phylum. These significant differences in microbiota composition between grazing- and feedlot- Angus beef may have an impact on the meat quality of Angus beef.

## Figures and Tables

**Figure 1 animals-11-03167-f001:**
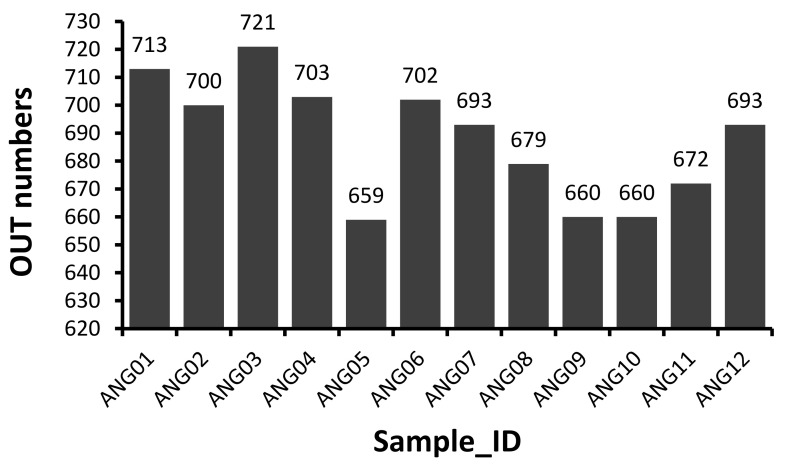
Distribution of 16S OTU number of each sample.

**Figure 2 animals-11-03167-f002:**
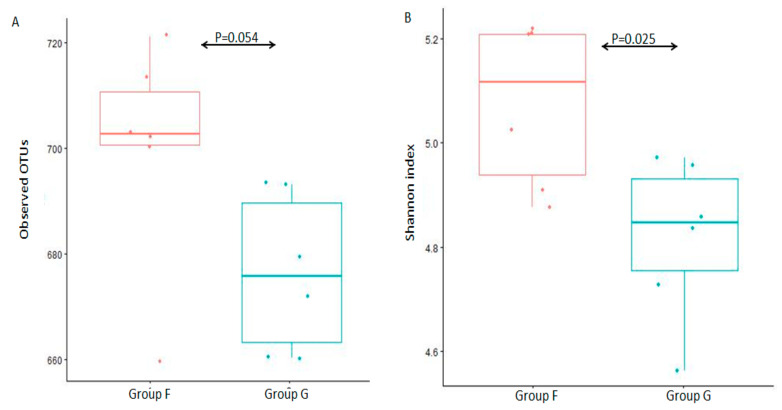
Box-plot representation of alpha diversity. Fecal microbiota were evaluated by the number of observed OTUs (**A**) and Shannon index (**B**) between group F and group G.

**Figure 3 animals-11-03167-f003:**
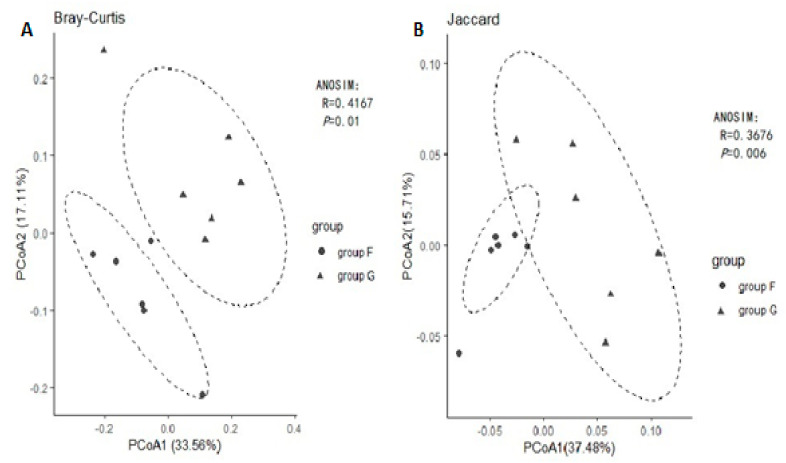
Principal Coordinates Analysis (PCoA) using Bray-Curtis distance (**A**) and Jaccard distance (**B**).

**Figure 4 animals-11-03167-f004:**
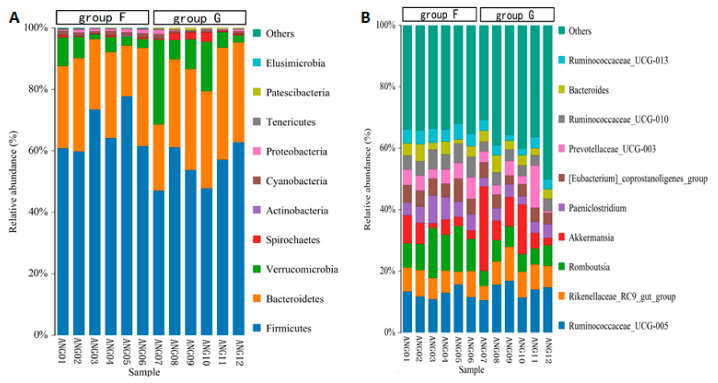
Relative abundance of fecal microbiota at the phylum level (**A**) and at the genus level (**B**).

## Data Availability

The data presented in this study are available on request from the corresponding author.

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
