# Peer review of "Composition of Fecal Microbiota in Grazing and Feedlot Angus Beef Cattle"

_animals, 2021, doi:10.3390/ani11113167_

Round 1
Reviewer 1 Report
Introduction
IN general OK, but the hypothesis of the authors should be made more clear.
M & M
2.1. Please delete, no interventions that could compromise animal welfare were made in this study. We must care about experimental animals, but we should not overreact.
2.2. Please explain the process for selection of animals and allocation into groups, i.e., all the details about randomisation etc.
Faeces collected from the inside: the correct expression is directly from the rectum
The manuscript needs significant improvement in English language, this and elsewhere are significant linguistic slips.
2.3. Please provide all the details of the PCR: primers and conditions in a single table, not in the text.
Results
Figures 1 and 2: please colorise
Discussion
The discussion focuses on the bacteriological – molecular findings per se and fails completely to address the underlying issue.
The authors should re-write the discussion from scratch, reducing the technical and molecular points and presenting the potential consequences of their findings for the digestive physiology of the cattle. Also, recommendations to go into the field of applied nutrition should be devised.
As it is now, the manuscript is not a scientific paper, but a very good technical report.
The work is done with the aim to improve the applied sciences, not just presenting the molecular results.
Finally, an extensive and thorough revision of the language should be performed.
Author Response
2.1. Please delete, no interventions that could compromise animal welfare were made in this study. We must care about experimental animals, but we should not overreact.
Re: We deleted the Ethics Statement in the manuscript.
2.2. Please explain the process for selection of animals and allocation into groups, i.e., all the details about randomisation etc.
Re: Six cattle (ANG1~6, group F) were randomly selected from a herd, which were kept in one indoor feedlot and fed with total mixed ration diet based on corn silage according to the Nation Research Council (NRC). Another six cattle (ANG7~12, group G) were selected from a herd, which grazed in a wild grassland and supplemented only mineral salt.
Faeces collected from the inside: the correct expression is directly from the rectum
Re: Samples were collected directly from the rectum of each cattle using a disposable glove,
The manuscript needs significant improvement in English language, this and elsewhere are significant linguistic slips.
Re: We checked the spelling and grammars through the manuscript.
2.3. Please provide all the details of the PCR: primers and conditions in a single table, not in the text.
Re: We used the universal primers for amplifying the V3-V4 regions of bacterial 16S rRNA gene. The primer is 338F and 806R.
Results
Figures 1 and 2: please colorise
Re: Figure 1 and 2 were coloried.
Discussion
The discussion focuses on the bacteriological – molecular findings per se and fails completely to address the underlying issue.
Re: Different fecal microorganisms only explained a part of the difference in nutrient absorption of beef cattle and its effect on beef quality.
The authors should re-write the discussion from scratch, reducing the technical and molecular points and presenting the potential consequences of their findings for the digestive physiology of the cattle. Also, recommendations to go into the field of applied nutrition should be devised.As it is now, the manuscript is not a scientific paper, but a very good technical report.The work is done with the aim to improve the applied sciences, not just presenting the molecular results.Finally, an extensive and thorough revision of the language should be performed.
Re: The objective of this study is to investigate the difference of bovine fecal microbiota between grazing and feedlot Angus cattle. The results may provied the possiblility to improve the quality of beef by altering cattle gut microorganism composition.
Fecal microbiota transplantation (FMT) is an increasingly popular therapy to the treatment of diseases such as metabolic syndrome, diabetes, Crohn's disease, Parkinson's disease, multiple sclerosis, psoriasis, anorexia nervosa or Alzheimer disease in human. Recently, FMT is successfully applied to ameliorate diarrhea and improv growth performance in pre-weaning calves. In this study, the relative abundance of Firmicutes, Cyanobacteria, Elusimicrobia and Patescibacteria was significantly difference between feedlot and grazing Angus beefs, which may provied the possiblility to improve the quality of beef by FMT and Microecologic agent.
We checked the spelling and grammars through the manuscript.

Reviewer 2 Report
The submitted for review paper compares the composition of the fecal microbiota of Angus cattle grazed under pasture and forage systems. The paper is interesting because of the subject matter.
However, I have some essential comments (see attachment).
In my opinion, the paper requires a major revision.

Author Response
Line 42-44: This statement does not fit the context, unclear, please change.
Re: We changed the sentence and find the right literature. “Because modern people very concerns about energy and nutritional balance, consumers are increasingly favoring grazing-fed beef”
Line 46-50: This relates to the rumen microbiome. Please provide information on the role the colonic microbiome.
Re: The fecal microbiome could be used as a proxy for the rumen microbiome due to its accessibility.
Line 82-85: Why group "0" (before the start of the experiment) was not included. No group 0 results in missing information about the state of the microbiota before the experiment.
Re: The objective of this study is to investigate the difference of bovine fecal microbiota between grazing and feedlot Angus cattle. The cattle always stay the same state. It doesn't matter the stage before the experiment.
Line 173: Not only in China. Delete
Re: modified.
Line 182: The citation is a reference to poultry. Please find another one.
Re: This literature provides a good example for the flavor of meat affected by gut microbiota in farm animal. We changed the “beef” to meat.
Line 199-210: Please change this section and find information about ruminants. This article is about ruminants and comparing with other animals with other types of digestive systems can be misleading.
Re: There is few report on the function of these phylum in the gut of ruminants, while their function reported in humans and mice is consistent with the phenomenon observed in this study. So, we used this references.
Line 217-221: Please change this section and find information about ruminants. This article is about ruminants and comparing with other animals with other types of digestive systems can be misleading
Re: There is few report on the function of these phylum in the gut of ruminants, while their function reported in humans is consistent with the phenomenon observed in this study. So, we used this references.
In my opinion, the paper requires a major revision.
Re: We revised this paper according to revewers’ advises.

Round 2
Reviewer 1 Report
The manuscript has been improved after the changes of the authors.
However, language problems still remain a significant problem and the manuscript needs extensive linguistic improvements before acceptance.
The necessary changes are extensive and in all aspects of language. As it is at the moment, it is difficult for readers to understand the ideas in the manuscript.
Author Response
Point 1: The manuscript has been improved after the changes of the authors. However, language problems still remain a significant problem and the manuscript needs extensive linguistic improvements before acceptance. The necessary changes are extensive and in all aspects of language. As it is at the moment, it is difficult for readers to understand the ideas in the manuscript.
Response 1: We have carried out sentence-by-sentence grammatical error correction and logical carding for the manuscript.
Reviewer 2 Report
I have no comments. In my opinion, the article is suitable for publication in its present form
Author Response
Point 1: I have no comments. In my opinion, the article is suitable for publication in its present form.
Response 1: We appreciated your advises.